# Meropenem Pharmacokinetics and Target Attainment in Critically Ill Patients Are Not Affected by Extracorporeal Membrane Oxygenation: A Matched Cohort Analysis

**DOI:** 10.3390/microorganisms9061310

**Published:** 2021-06-16

**Authors:** Matthias Gijsen, Erwin Dreesen, Pieter Annaert, Johan Nicolai, Yves Debaveye, Joost Wauters, Isabel Spriet

**Affiliations:** 1Clinical Pharmacology and Pharmacotherapy, Department of Pharmaceutical and Pharmacological Sciences, KU Leuven, 3000 Leuven, Belgium; erwin.dreesen@kuleuven.be (E.D.); isabel.spriet@uzleuven.be (I.S.); 2Pharmacy Department, UZ Leuven, 3000 Leuven, Belgium; 3Drug Delivery and Disposition, Department of Pharmaceutical and Pharmacological Sciences, KU Leuven, 3000 Leuven, Belgium; pieter.annaert@kuleuven.be (P.A.); johan.nicolai@ucb.com (J.N.); 4BioNotus, 2845 Niel, Belgium; 5Development Science, UCB Biopharma SRL, 1420 Braine-l’Alleud, Belgium; 6Laboratory for Intensive Care Medicine, Department of Cellular and Molecular Medicine, KU Leuven, 3000 Leuven, Belgium; yves.debaveye@uzleuven.be; 7Medical Intensive Care Unit, UZ Leuven, 3000 Leuven, Belgium; joost.wauters@uzleuven.be; 8Laboratory for Clinical Infectious and Inflammatory Diseases, Department of Microbiology, Immunology and Transplantation, KU Leuven, 3000 Leuven, Belgium

**Keywords:** meropenem, intensive care, pharmacokinetics, population pharmacokinetics, extracorporeal membrane oxygenation, target attainment

## Abstract

Existing evidence is inconclusive whether meropenem dosing should be adjusted in patients receiving extracorporeal membrane oxygenation (ECMO). Therefore, the aim of this observational matched cohort study was to evaluate the effect of ECMO on pharmacokinetic (PK) variability and target attainment (TA) of meropenem. Patients admitted to the intensive care unit (ICU) simultaneously treated with meropenem and ECMO were eligible. Patients were matched 1:1, based on renal function and body weight, with non-ECMO ICU patients. Meropenem blood sampling was performed over one or two dosing intervals. Population PK modelling was performed using NONMEM7.5. TA was defined as free meropenem concentrations >2 or 8 mg/L (i.e., 1 or 4× minimal inhibitory concentration, respectively) throughout the whole dosing interval. In total, 25 patients were included, contributing 27 dosing intervals. The overall TA was 56% and 26% for the 2 mg/L and 8 mg/L target, respectively. Population PK modelling identified estimated glomerular filtration rate according to the Chronic Kidney Disease Epidemiology equation and body weight, but not ECMO, as significant predictors. In conclusion, TA of meropenem was confirmed to be poor under standard dosing in critically ill patients but was not found to be influenced by ECMO. Future studies should focus on applying dose optimisation strategies for meropenem based on renal function, regardless of ECMO.

## 1. Introduction

Since the publication of the CESAR trial, venovenous extracorporeal membrane oxygenation (vvECMO) has been used increasingly in adults with severe acute respiratory distress syndrome [1]. Similarly, there is increasing evidence supporting the use of venoarterial ECMO (vaECMO) in patients with severe cardiac failure [2]. Both vvECMO and vaECMO have been associated with increased survival rates when applied in specialised ECMO centres [3].

Infectious complications are commonplace in patients supported by ECMO and have been identified both as indication [4,5,6,7] and complication of ECMO [8,9,10]. High rates of nosocomial infections with multidrug resistant bacteria, associated with increased mortality, have been reported in these patients [8,9,10,11]. Hence, it is no surprise that patients undergoing ECMO are often treated with broad-spectrum antimicrobial agents, such as meropenem.

Unfortunately, ECMO has the potential of further increasing the variability in antimicrobial exposure and pharmacokinetics (PK) in critically ill patients. Due to hemodilution and drug sequestration, ECMO has the potential to further increase the volume of distribution in critically ill patients. Similarly, ECMO has been associated with augmented cardiac output, which could lead to increased drug clearance [12]. Therefore, several studies have investigated the effect of ECMO on antimicrobial PK, and the need for dose adjustments in these patients. Drug sequestration has been shown for lipophilic and highly protein-bound drugs [12,13]. However, it remains unclear whether ECMO has a significant effect on the PK of hydrophilic antimicrobials such as beta-lactams. An ex vivo study suggested substantial sequestration of meropenem in the ECMO circuit [14]. Currently, limited clinical evidence suggests that PK alterations are rather the consequence of critical illness than ECMO [15,16]. Population PK (popPK) studies performed in critically ill non-ECMO patients found renal function and (to a lesser extent) body weight to be significant predictors for meropenem PK alterations [17]. A recent review recommended beta-lactam dosing in ECMO patients to align with dosing strategies used for critically ill patients not supported by ECMO [18]. In contrast, the largest therapeutic drug monitoring (TDM) study performed in ECMO patients until now, published last year, reported significantly lower meropenem plasma concentrations in ECMO patients [19]. As a result, it is still unclear whether meropenem PK is significantly affected by ECMO, and whether meropenem dosing should be adjusted in these patients.

The aim of the present study is to investigate the effect of ECMO on PK variability and pharmacokinetic/pharmacodynamic (PK/PD) target attainment of meropenem in critically ill patients. Therefore, a cohort of ECMO patients was matched with a cohort of non-ECMO critically ill patients.

## 2. Materials and Methods

### 2.1. Setting, Study Design and Population

We performed a prospective single-centre observational matched cohort study on the ICUs of a tertiary-care academic hospital (University Hospitals Leuven, Leuven, Belgium) between October 2013 and October 2017. The study was conducted in accordance with the Declaration of Helsinki, and the study protocol was approved by the Ethics Committee Research UZ/KU Leuven (S54511). Written informed consent was obtained from the patient or the closest relative before inclusion in this study. In this study, PK/PD target attainment of meropenem was investigated in two cohorts of critically ill patients, ECMO and non-ECMO patients. For the ECMO cohort, all adult patients admitted to the ICU, supported by ECMO and treated with meropenem, were screened for eligibility. Pregnant women and patients with a do-not-resuscitate order were excluded. The non-ECMO cohort focused on septic patients with preserved renal function. For this cohort, all adult patients admitted to the ICU, treated with meropenem and having severe sepsis or septic shock (as defined according to the definitions applicable at the time of the study; cf. Appendix A) [20,21] at the start of meropenem therapy were screened. Exclusion criteria for this cohort were pregnancy, do-not-resuscitate order, ECMO, renal replacement therapy (RRT), estimated glomerular filtration rate according to the Chronic Kidney Disease Epidemiology Collaboration equation (eGFR_CKD-EPI_) < 70 mL/min/1.73 m^2^ on the day of PK sampling.

### 2.2. Study Protocol

For both the ECMO and non-ECMO cohort, the same study protocol was applied. Meropenem dosing in all patients was at the discretion of the treating physician. Meropenem was administered as a 50 mL infusion over 30 min. Blood samples were taken over one dosing interval at −15 (pre-dose), 30 (at time of infusion stop), 120, and 240 min after the start of the infusion, and 15 min before the start of the next infusion (i.e., at trough). Depending on practical feasibility, sampling was performed on an early day (day 2 +/−1) and/or a late day (day 4 or later) of meropenem therapy. Blood samples were collected in lithium heparinised tubes (4 mL), immediately refrigerated (4 °C), and the plasma obtained after centrifugation was stored at −20 °C within 2 h post sampling [22]. During the first year of the study, plasma samples from nine patients were stored at −20 °C until quantification. However, new insights revealed meropenem degradation at −20 °C [23,24]. Subsequently, from the second year of the study on, within 24 h after collection, all plasma samples were transferred at −80 °C until quantification. Meropenem concentrations quantified from blood samples that had been stored at −20 °C for maximum one year were corrected using a validated model for degradation of meropenem at −20 °C [25]. The total meropenem concentrations were measured and reported as free concentrations, as meropenem shows negligible plasma protein binding (i.e., 2%) [26]. Meropenem plasma concentrations were quantified using a validated ultra-performance liquid chromatography method coupled with tandem mass spectrometry. The bioanalytical method is provided in Appendix A.

### 2.3. Matching Procedure

We decided to match per dosing interval and not per patient, because important intra-individual variability in meropenem PK has been shown in critically ill patients [27]. Dosing intervals from ECMO patients were matched 1:1 with dosing intervals from non-ECMO ICU patients. The renal function and (actual) body weight were selected as matching criteria, as these were previously identified as predictors for meropenem exposure in several non-ECMO meropenem popPK models [17]. Matching was performed based on renal function on the day of PK sampling (defined as 24-h measured urinary creatinine clearance (mCrCL_24h_) if available, else eGFR_CKD-EPI_) and body weight within ± 20% of the value related to the ECMO dosing interval, or when not possible, the closest match.

### 2.4. PK/PD Target Attainment

PK/PD target attainment of beta-lactams is commonly assessed according to the percentage of the dosing interval that the free drug concentration exceeds the minimal inhibitory concentration (MIC) (i.e., %*f*T_>MIC_). For beta-lactams, a target of 100% *f*T_>MIC_ has been recommended in critically ill patients to maximize bacterial killing and to suppress resistant mutants [28]. Furthermore, 100% *f*T_>4×MIC_ has been suggested to achieve maximal clinical outcome [29,30]. Both targets have been recommended when using beta-lactams in the ICU [31,32,33]. Therefore, in the present study, PK/PD target attainment was defined as a free concentration above the MIC or 4-fold the MIC throughout the whole dosing interval (i.e., 100% *f*T_>MIC_ and 100% *f*T_>4×MIC_, respectively). The clinical MIC breakpoint for susceptibility of *Enterobacterales* to meropenem of 2 mg/L—recommended by the European Committee on Antimicrobial Susceptibility Testing (http://www.eucast.org/clinical_breakpoints (accessed on 24 March 2021))—was used. Hence, PK/PD target attainment is defined as when the meropenem free concentration exceeds 2 or 8 mg/L throughout the whole dosing interval, corresponding to 100% *f*T_>MIC_ or 100% *f*T_>4×MIC_, respectively.

### 2.5. Population PK Modelling

A popPK model was built using NONMEM (version 7.5; ICON Development Solutions, Gaitherburg, MD, USA). All procedures were executed using the Perl-speaks-NONMEM (PsN; version 5.0.0) toolkit on the Pirana modelling workbench (version 2.9.9; Certara, Inc. Princeton, NJ, USA). The first-order conditional estimation method with interaction (FOCE-I) was used to estimate model parameters with the differential equation solver ADVAN 13. Model selection and evaluation were based on the precision of the parameter estimates, the condition number, goodness of fit (GOF) diagnostics, visual predictive checks (VPCs), and nonparametric bootstrapping. The likelihood ratio test was used to compare GOF of two nested models, assuming that the objective function value (OFV) follows a χ^2^ distribution (ΔOFV of 3.84 point indicates significance at the 5% level for models differing in one parameter). The Akaike information criterion, defined as OFV + 2 × k, with k the number of estimated fixed and random effects parameters, was used to compare the GOF of models that were not nested.

Structural models with one and two compartments were evaluated. Interindividual variability (IIV) terms were tested, assuming random, log-normal parameter distributions. Next, patient covariates were sought to explain the IIV using a stepwise covariate modelling (SCM) procedure with forward addition (α = 0.05) and backward elimination (α = 0.01). Parameter-covariate relations were included as power function for continuous covariates (mCrCL_24h_, estimated creatinine clearance using the Cockcroft–Gault formula (eCrCL_CG_), eGFR using the Modification of Diet in Renal Diseases equation (eGFR_MDRD_), the eGFR_CKD-EPI_, body weight, body mass index, ideal body weight, adjusted body weight, age, acute physiology and chronic health evaluation II (APACHE II) score, sequential organ failure assessment (SOFA) score at ICU admission and day of PK sampling, serum albumin, total protein, total bilirubin, and fluid balance). Categorical covariates (sex, need for ECMO, mechanical ventilation, and vasopressor therapy) were included as a shift in the typical value from the most common category.

The NONMEM control stream of the final popPK model is available in the Appendix A.

### 2.6. Statistical Analysis

The results are expressed as median (interquartile range) or median (range), as indicated. All analyses other than the popPK analyses were performed in R (version 3.5.1 or higher, R Core Team, Vienna, Austria) in the RStudio integrated development environment (version 1.3; RStudio, Inc., Boston, MA, USA) using the tidyverse [34] collection of packages, nlme [35] and ggplot2 [36].

## 3. Results

### 3.1. Patient Characteristics and Matching

Fourteen ECMO patients were included, contributing 15 dosing intervals (Appendix A). Only two patients were supported by vaECMO, and three ECMO dosing intervals were included during continuous renal replacement therapy (CRRT). Details of the ECMO therapy are reported in the Appendix A. As none of the patients in the non-ECMO cohort received CRRT, and renal estimators are inappropriate in this setting, only non-CRRT ECMO dosing intervals were matched with controls. These 12 ECMO dosing intervals were matched with controls (i.e., non-ECMO) within 20% for body weight and renal function, except for 1 ECMO dosing interval (mCrCl_24h_ 13 vs. 29 mL/min/1.73 m^2^). For the non-ECMO dosing intervals, matching dosing intervals were selected from 70 dosing intervals available (Appendix A). Clinical characteristics are shown in Table 1. There were no significant differences between ECMOs and controls, except for age (*p* = 0.0008) and SOFA score on the day of PK sampling (*p* = 0.006). For one matched pair, eGFR_CKD-EPI_ was used for matching and reporting, as mCrCl_24h_ was not available for one ECMO patient.

Infection focus was mostly respiratory (100% in ECMO and 50% in non-ECMO cohort). The majority of PK samplings were performed on an early day during meropenem therapy (8 ECMO and 10 non-ECMO dosing intervals). For most dosing intervals, meropenem was administered as 1 g q8h (10 ECMO and 8 non-ECMO dosing intervals). The alternative dosing regimen was 2 g q8h, except for 1 ECMO patient who received 1 g q12h. As a result, there was no significant difference in meropenem dose administered in the ECMO vs. the non-ECMO cohort.

### 3.2. PK/PD Target Attainment

Median (range) free meropenem trough plasma concentrations were 4.4 (0.13–18.8) mg/L and 2.7 (0.39–33.9) mg/L for the ECMO and non-ECMO dosing intervals, respectively. As illustrated in Figure 1, there is no significant difference in trough concentrations (*p* = 0.643) between both cohorts. Across both patient cohorts, 100% *f*T_>MIC_ and 100% *f*T_>4×MIC_ were successfully attained in 56% and 26% of all dosing intervals, respectively. For both PK/PD targets, attainment was not significantly different between the ECMO vs. the non-ECMO cohort. In the ECMO cohort, 100% *f*T_>MIC_ and 100% *f*T_>4×MIC_ were successfully attained in 8 (53%) and 4 (27%) of the 15 dosing intervals, respectively. When considering only non-CRRT dosing intervals, 100% *f*T_>MIC_ and 100% *f*T_>4×MIC_ were successfully attained in 5 (42%) and 3 (25%) of the 12 ECMO dosing intervals, respectively. In the non-ECMO cohort, 100% *f*T_>MIC_ and 100% *f*T_>4×MIC_ were successfully attained in 7 (58%) and 3 (25%) of the 12 dosing intervals, respectively.

### 3.3. Population Pharmacokinetic Modelling

The 3 ECMO dosing intervals on CRRT were excluded from the popPK analysis. Hence, popPK modelling was performed only with dosing intervals for which matching based on renal function was possible (i.e., 24 dosing intervals of 22 patients).

The meropenem concentrations were best described by a two-compartment model with linear elimination. The parameters of the final popPK model are shown in Table 2. GOF diagnostic plots (Appendix A), and a VPC (Figure 2) both demonstrate an adequate fit of the model to the data. PK profiles showing observed and individual predicted meropenem concentrations are provided in Appendix A.

The eGFR_CKD-EPI_ was found to explain 20.5% (coefficient of variation) of the interindividual variability on meropenem clearance (ΔOFV = 26.7 points). Meropenem clearance (CL) was proportional to eGFR_CKD-EPI_ as follows:CL_i_ = 14.7 × (BW_i_/70)^0.75^ × (eGFR_CKD-EPI,i_/105)^1.29^(1)
where CL_i_ is individual meropenem clearance, BW_i_ is individual body weight and eGFR_CKD-EPI,i_ is individual eGFR_CKD-EPI_.

Mean estimated meropenem clearance and volume of distribution in the central compartment were similar between non-ECMO and ECMO patients (clearance: 13.7 ± 8.44 vs. 17.4 ± 14.8 L/h, *p* = 0.462, and volume of distribution in the central compartment: 31.3 ± 18.3 vs. 29.7 ± 19.2 L, *p* = 0.837). As a result, ECMO was not retained as a significant predictor in the popPK analysis.

Random unexplained interindividual variability in meropenem clearance and volume of distribution remained large (46.8% and 61.6%, respectively). Consequently, terminal half-lives varied within a wide range (1.3 to 10.2 h).

## 4. Discussion

In this matched cohort study, we showed that eGFR_CKD-EPI_ and body weight were the only variables significantly influencing meropenem PK in critically ill patients with severe sepsis or septic shock. ECMO was not retained in the popPK model; hence, it was not found to influence meropenem PK. PK/PD target attainment was equally poor in both the ECMO and non-ECMO cohort, illustrating the need to apply dose optimisation strategies for meropenem in critically ill patients regardless of ECMO.

No significant difference in PK parameters and target attainment was found in ECMO vs. non-ECMO patients. These results are in line with Donadello et al. who concluded that achievement of target concentrations for meropenem and piperacillin-tazobactam does not appear to be influenced by ECMO [16]. In vitro ECMO experiments showed no significant loss of meropenem after 24 h [37]. Hence, it appears that there is no significant sequestration of meropenem in the ECMO circuit. Recent studies have shown that lipophilic drugs and drugs with high plasma protein binding are prone to sequestration within the ECMO circuit [38,39]. Therefore, our results are not surprising, as meropenem is a hydrophilic drug (log P −0.6) [40] and shows low plasma protein binding (2%) [26]. In contrast, a recent unmatched cohort study found ECMO treatment to be associated with lower serum concentrations of meropenem [19]. They reported that sequestration of meropenem in the ECMO circuit is a possible reason for this. Interestingly, meropenem was administered as a continuous infusion in their study and meropenem concentrations were relatively high, with median concentrations of 15 vs. 17.8 mg/L in ECMO and non-ECMO patients, respectively. Although being statistically different, the clinical relevance of this relatively small difference is debatable, especially with concentrations that remain well above 100% *f*T_>MIC_ and 100% *f*T_>4×MIC_ (i.e., 2 and 8 mg/L, respectively).

Distribution volumes found in the present study are similar to those reported by Shekar et al. in their matched ECMO cohort study [41]. However, meropenem CL is substantially higher in our study (14.6 L/h vs. 5.4 L/h). This is not surprising as the popPK modelling in our study did not include any dosing intervals associated with CRRT in contrast to Shekar et al. (CRRT in 10/21 patients) [41]. Meropenem CL has been shown to be lower in patients receiving CRRT [27]. Moreover, in the present study, most patients showed preserved or even increased renal function. This can potentially be attributed to the exclusion criteria applied to the non-ECMO cohort (i.e., RRT and eGFR_CKD-EPI_ < 70 mL/min/1.73 m^2^). Nevertheless, renal function was not significantly different between both cohorts, thus suggesting adequate matching. Meropenem CL found in the present study is similar to values reported in several popPK studies in critically ill patients not treated with ECMO or RRT [17]. It should be noted that there was still a substantial residual interindividual variability in meropenem clearance and volume of distribution. This is in accordance with other popPK models developed in ECMO [41] and non-ECMO patients [17]. Especially for the volume of distribution, the interindividual variability remained large. This might be due to unmeasured covariates. Interestingly, capillary leak has been suggested to increase the volume of distribution of hydrophilic drugs in septic ICU patients [42]. However, until now, there is no readily available biomarker or technique to quantify a capillary leak, which might in turn be used to explain the variability in the volume of distribution. Therefore, future studies might focus on the assessment of capillary leak, and its association with the volume of distribution of meropenem (or other hydrophilic antimicrobials).

In the present study, the PK/PD target attainment was low, regardless of ECMO, as illustrated by an overall target attainment of 56% and 26% for 100% *f*T_>MIC_ and 100% *f*T_>4×MIC_, respectively. Renal function was the major driver for increased meropenem clearance and consequently for the high incidence of target non-attainment. This confirms similar findings from previous smaller ECMO studies. Hanberg et al. warned for an unacceptable lack of target attainment in ECMO patients, especially in those with increased renal function [15]. In a matched cohort study examining the effects of ECMO and renal replacement therapy, CrCL_CG_ and renal replacement therapy were correlated with meropenem clearance [41]. In the previously mentioned unmatched cohort study of Kuhn et al., PK/PD targets were attained in >90% of all TDM measurements for meropenem [19]. This stands in great contrast to all previous studies. This could be explained by two major reasons. First, serial TDM measurements per patient were included in their study, but once dose adaptations were needed (due to target non-attainment), further TDM measurements were excluded from the analysis. Hence, a selection bias towards TDM measurements within predefined targets is very likely. Second, many patients (53% vs. 20% of ECMO patients) received CRRT. As a result, Kuhn et al. found CRRT, and not mCrCL_24h_, to be associated with increased meropenem concentrations. This suggests that mCrCL_24h_ (value not reported) was relatively low in their study, hence increasing the probability of target attainment. Although dosing was similar to our study (67% vs. 63% received 3 g/day), it should be mentioned that meropenem was administered via continuous infusion in their study, which has been shown to optimise T_>MIC_ [43]. Importantly, in the present study, most patients showed preserved or increased renal function, which has been shown to be the major reason for failure of target attainment of meropenem in both ECMO [15,16,41] and non-ECMO patients [17]. In critically ill patients not supported by ECMO, similar rates of PK/PD target attainment (i.e., 48% and 21% for 100% *f*T_>MIC_ and 100% *f*T_>4×MIC_, respectively) have been reported as the rates found in the present study [44].

This study has several limitations. First, 3 CRRT ECMO dosing intervals were included, which could not be matched with controls as no patients in the non-ECMO cohort received CRRT. As a result, no conclusions can be drawn for CRRT ECMO patients. Nevertheless, descriptive statistics did not find any significant difference in target attainment during ECMO vs. non-ECMO dosing intervals when ECMO patients on CRRT were included. Second, one dosing interval was not matched within 20% of renal function. Albeit, the absolute difference was relatively small (i.e., mCrCl_24h_ 13 vs. 29 mL/min/1.73 m^2^), with mCrCl_24h_ falling below 30 mL/min/1.73 m^2^ for both dosing intervals. Furthermore, renal function was similar in both cohorts, suggesting adequate matching. Only age and SOFA score on the day of PK sampling differed in both cohorts, which is a natural consequence of ECMO being applied preferentially in younger patients, and ECMO patients being per definition more severely ill. This probably did not influence our conclusions as neither age, nor SOFA were retained as significant predictors of meropenem PK. Third, at the start of the study, blood samples were stored at −20 °C for maximum one year, whereas significant meropenem degradation has been reported after approx. 3–5 months at −20 °C [25]. Meropenem concentrations were corrected for this degradation using a robust and validated degradation model. A sensitivity analysis, excluding samples corrected for storage at −20 °C, revealed similar target attainment. Additionally, concentrations that had been corrected for storage at −20 °C were not significantly different from concentrations measured in samples that were stored only at −80 °C (i.e., not needing correction for storage at −20 °C). Fourth, no individual MICs were available, hence we evaluated target attainment according to a worst-case scenario using the clinical MIC breakpoint. This is a strategy reflecting empirical therapy in which the pathogen and its MIC are usually unknown. Lastly, eGFR_CKD-EPI_ and not mCrCL_24h_, which is known to be more appropriate in ICU patients [45], was retained as renal function marker in the popPK model. This is probably due to missing mCrCL_24h_ values in two patients. Nevertheless, when mCrCL_24h_ is not available, eGFR_CKD-EPI_ has been shown to be the best alternative estimator of renal function [45].

This study also has several strengths. First, this is a matched cohort study, which allows a robust analysis of the influence of ECMO on meropenem PK and target attainment, despite the limited sample size. Second, PK sampling covered the first five days of ECMO therapy, including the first 24 h. If ECMO has any effect on meropenem PK, this would probably be most significant during the first days of ECMO. Hence, it is unlikely that significant PK alteration from ECMO was missed in this study. Lastly, this is the largest popPK modelling study in ECMO patients to date. Although several studies have described meropenem PK in ECMO patients before, these studies included fewer ECMO patients [15,41], or performed sparse sampling without popPK modelling [16,19].

In conclusion, this study supports the hypothesis that failure of PK/PD target attainment for meropenem is due to critical illness rather than due to ECMO. In critically ill patients, the use of standard dosing of meropenem resulted in poor target attainment which was not found to be influenced by ECMO. As suggested in a recent review [18], beta-lactam (in this case, meropenem) dosing in ECMO patients should indeed generally align with the recommended dosing strategies for critically ill non-ECMO patients. Future studies should focus on applying dose optimisation strategies for meropenem based on renal function, regardless of ECMO.

## Figures and Tables

**Figure 1 microorganisms-09-01310-f001:**
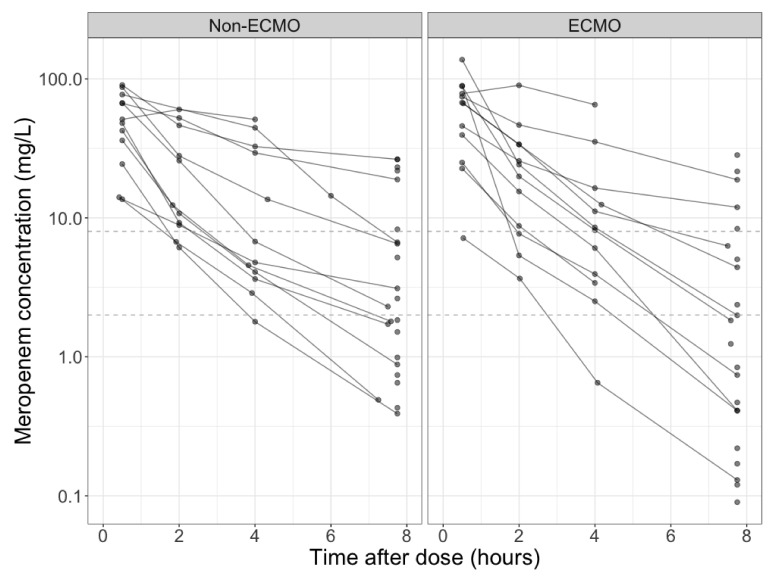
Observed free meropenem plasma concentrations vs. time relative to start infusion in non-ECMO (left) and ECMO patients (right). The dashed grey lines represent the 2 mg/L and 8 mg/L PK/PD targets.

**Figure 2 microorganisms-09-01310-f002:**
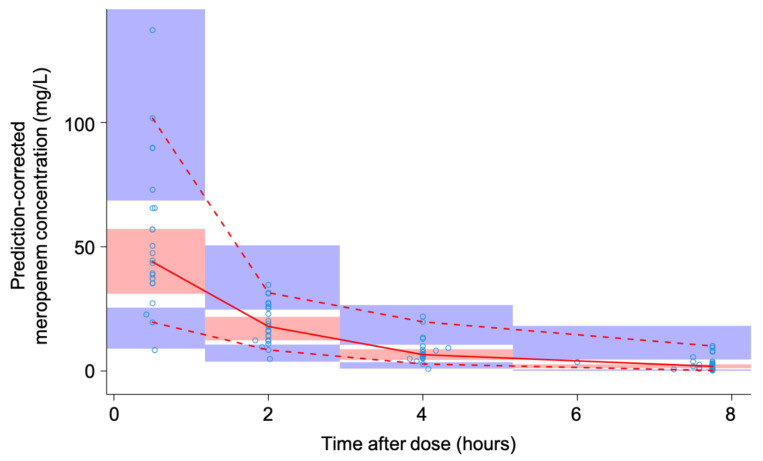
Prediction-corrected visual predictive check of the final population pharmacokinetic model. Blue open circles represent the observed free meropenem plasma concentrations. The solid red line is the median of the observed data. The dashed red lines are the 2.5th and 97.5th percentile of the observed data. The red and blue shaded areas indicate the 95% prediction intervals of the median and the 2.5th and 97.5th percentile, respectively, of the simulated data (*n* = 1000).

**Table 1 microorganisms-09-01310-t001:** Clinical characteristics.

**At Baseline (per Patient)**	**ECMO (*n* = 14)**	**non-ECMO (*n* = 11)**	***p-*Value**
Male, *n* (%)	9 (64)	7 (64)	0.7
Age, median [IQR], years	47 [39;56]	68 [63;71]	0.0008 *
Body weight, median [IQR], kg	84 [54.6;90]	70 [61;77.5]	0.337
SOFA score on admission, median [IQR]	9 [6;12]	8 [6;11]	0.62
APACHE II score, median [IQR]	21 [18;23]	21 [16;27]	0.1
ICU mortality, *n* (%)	2 (14)	3 (27)	0.628
Venoarterial ECMO, *n* (%)	2 (14)	NA	NA
**Dosing interval (on the day of PK sampling)**	**ECMO (*n* = 15)**	**non-ECMO (*n* = 12)**	
Days meropenem therapy until measurement, median [IQR]	2.7 [1.3;5]	1.5 [1;1.7]	0.086
Days ECMO therapy until measurement, median [IQR]	2.6 [1.2;4.4]	NA	NA
CRRT, *n* (%)	3 (20)	0 (0)	0.231
mCrCL_24h_, median [IQR], mg/dL ^a^	57 [47;120]	70 [46;112]	0.853
eGFR_CKD-EPI_, median [IQR], mL/min/1.73 m^2^ ^b^	106 [80;117]	97 [92;106]	0.544
Fluid balance, median [IQR]; mL	626 [−740;1193]	568 [−163;976]	0.942
SOFA score, median [IQR]	11 [8;14]	8 [5;9]	0.006 *
Daily meropenem dose, median [IQR]	3000 [3000;6000]	3000 [3000;6000]	0.574
Mechanical ventilation, *n* (%)	14 (93)	9 (75)	0.294
Vasopressor therapy, *n* (%)	9 (60)	7 (64)	1

^a^ mCrCL_24h_ was not available for 4 ECMO dosing intervals (of which 3 were on CRRT); ^b^ eGFR_CKD-EPI_ was not available for 1 ECMO dosing interval. APACHE: Acute Physiology and Chronic Health Evaluation, mCrCl_24h_: 24-h measured urinary creatinine clearance, CRRT: continuous renal replacement therapy, ECMO: extracorporeal membrane oxygenation, eGFR_CKD-EPI_: estimated glomerular filtration rate according to the Chronic Kidney Disease Epidemiology Collaboration equation, ICU: intensive care unit, IQR: interquartile range, PK: pharmacokinetic, SOFA: Sequential Organ Failure Assessment; * *p* < 0.05.

**Table 2 microorganisms-09-01310-t002:** Parameter estimates of the population pharmacokinetic model.

	Final Model	(OFV 437.6—CN 336.8)	Bootstrapped Estimate ^a^	
Parameter	Estimate	(%RSE) [Shrinkage]	Median	[95%CI]
Fixed effects				
CL (L/h)	14.7	(11)	14.6	[11.8–18.1]
BWT on CL	0.75	FIX		FIX
eGFR_CKD-EPI_ on CL	1.29	(16)	1.31	[1.03–2.85]
Vc (L)	25.6	(15)	24.2	[16.2–32.9]
BWT on V_c_	1	FIX		FIX
Q (L/h)	5.51	(29)	6.14	[2.98–37.5]
BWT on Q	0.75	FIX		FIX
Vp (L)	8.02	(23)	8.75	[5.54–17.94]
BWT on V_p_	1	FIX		FIX
Random effects				
IIV on CL (%CV)	46.8	(29) [1]	43.3	[24.4–68.2]
IIV on V_c_ (%CV)	61.6	(22) [8]	43.3	[−15.8–68.9]
Corr(IIV on CL-V_c_) (%CV)	70.4	(35)	62.0	[28.4–87.1]
Residual variability				
Proportional error (%CV)	28.8	(13.3) [15]	28.2	[21.2–36.3]

^a^ A total of 942 out of 1000 (94.2%) successfully minimised runs were used to calculate the median values of the parameter estimates and the 2.5th and 97.5th percentiles, defined by the lower and upper limits, respectively, of the 95% confidence interval for the population parameter estimates. BWT: body weight, CI: confidence interval, CL: clearance, CN: condition number, Corr: correlation, CV: coefficient of variation, eGFR_CKD-EPI_: estimated glomerular filtration rate according to the Chronic Kidney Disease Epidemiology Collaboration equation, IIV: interindividual variability, OFV: objective function value, Q: intercompartmental clearance, RSE: relative standard error (for random effects and residual variability reported on the approximate standard deviation scale; variance estimate/2), V_c_: volume of distribution in the central compartment, V_p_: volume of distribution in the peripheral compartment.

## Data Availability

The data presented in this study are available on request from the corresponding author. The data are not publicly available due to privacy and ethical reasons.

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
