# Peer review of "Meropenem Pharmacokinetics and Target Attainment in Critically Ill Patients Are Not Affected by Extracorporeal Membrane Oxygenation: A Matched Cohort Analysis"

_microorganisms, 2021, doi:10.3390/microorganisms9061310_

Round 1

Reviewer 1 Report

The authors made a meropenem (MEPM) pharmacokinetics (PK) model in severe patients with and without ECMO, and researched an impact of ECMO on PK parameters of MEPM. This study is particular interest because of increasing use of ECMO in COVID-19 catastrophe and should be considered for additional improvement.

Major

  1. Are conditions of ECMO same among included patients? Please describe the information of ECMO.
  2. Method: Blood samples were taken at several points except for trough (480 mins after an administration of MEPM). Moreover, the authors evaluated a target attainment of MEPM using 100% fT>MIC. It is impossible to assess whether the trough is over MIC.
  3. Method: What difference is between 100% fT>MIC and 100% fT> 4×MIC? Is 100% fT> 4×MIC the target for severe patients?
  4. Result: The authors should show a figure of MEPM concentration and time after dose for all included patients. Moreover, they need to display MICs of bacteria detected in the patients.

Minor

  1. Abstract: Please explain an abbreviation of eGFRCKD-EPI.
  2. Method: Please explain an abbreviation of fT>MIC.
  3. Method: Please describe how to assess the Akaike information criterion.
  4. Table 1: Please change from Control to non-ECMO. Moreover, please add the P value in each item.

Reviewer 2 Report

In this work, the authors reported an observational matched cohort study to evaluate the effect of extracorporeal membrane oxygenation on pharmacokinetic variability and target attainment of meropenem. This is an interesting work while the authors are being requested to go through the article to revise some issues. Following are my questions and comments for reworking the manuscript.
1. Keywords: “antibacterial dosing” is not an eligible keyword. Please delete it.
2. “Infectious complications are commonplace in patients supported by ECMO and have been identified both as indication and complication of ECMO” This statement is the key background of this work. More supporting references should be included other than Ref 4.
3. Line 52: It is unclear why ECMO has the potential of further increasing the variability in antimicrobial exposure and pharmacokinetics in critically ill patients?
4. For two enrolled cohorts of critically ill patients, ECMO 81 and non-ECMO patients patient, whether the prior dosing history of meropenem has been taken into account in this work?
5. Table 1: It seems there is a significant difference between ECMOs and controls in age. I am wondering whether this difference can affect the conclusion of this work?
6. To improve the readability, I recommend splitting Figure 1 into two to present the ECMO and non-ECMO groups respectively since too many lines were included in the current one.

Round 2

Reviewer 1 Report

The authors revised appropriately. No further correction is necessary.